# Laminated Flow-Cell Detector with Granulated Scintillator for the Detection of Tritiated Water

**Nile E. J. Dixon** [1] ⓘ, **Stephen D. Monk** [1] ⓘ, **James Graham** [2] ⓘ and **David Cheneler** [1,*] ⓘ

1 Engineering Department, Lancaster University, Lancaster LA1 4YW, UK; n.dixon@lancaster.ac.uk (N.E.J.D.); s.monk@lancaster.ac.uk (S.D.M.)
2 National Nuclear Laboratory, Central Laboratory, Sellafield CA20 1PG, UK; james.graham@uknnl.com
* Correspondence: d.cheneler@lancaster.ac.uk

**Simple Summary:** Tritium, a radioactive isotope of hydrogen, is incredibly difficult to detect at a distance, making conventional monitoring equipment like Geiger Müller detectors unsuitable. This is partly because tritium only emits low-energy beta radiation, which is easily absorbed by surrounding matter, including the mica window of a Geiger Müller tube, which typically can only measure higher energy radiation emitted from heavier isotopes (the limit is Carbon-14, which emits 156 keV beta). Scintillators are an alternative radiation-monitoring approach and are used to detect radiation by interacting with these betas, producing light that itself can be detected using high-sensitivity light sensors. This work tests a granulated scintillator that can be mixed directly with water-containing tritium and kept contained within a flow cell formed using heat lamination. Results show a relationship between powder fineness and detection rate, as well as an increased count rate when tritium is added to a manufactured laminated flow cell containing a granulated scintillator.

**Abstract:** Nuclear sites require regular measurements of the air, soil, and groundwater to ensure the safety of the surrounding environment from potentially hazardous levels of contamination. Although high-energy beta and gamma emitters can often be detected instantly using fixed dosimeters, the detection of low-energy beta emitters is a difficult challenge, especially in groundwater, as its radiation is easily self-absorbed by the surrounding medium. Therefore, it is common practice to sample groundwater and transfer it to a laboratory for analysis using Liquid Scintillation Counting. This work demonstrates a new detector design for the real-time monitoring of tritiated water, a weak beta emitter. This design utilizes a flow cell filled with a granulated scintillator to maximize the surface area of the sample. The cavity is made from plastic sheets, which allow rapid manufacture using readily available lamination sheets. A column of SiPMs in coincidence counting mode has been implemented to reduce noise and allow future extensions to the flow cell for greater detection rates while allowing the detector to fit within limited spaces such as groundwater monitoring boreholes. Using multiple concentrations of tritiated water, this detector has been validated and calibrated, obtaining a minimum detection activity of $26.356 \pm 0.889$ Bq/mL for a 1-day counting period.

**Keywords:** flow-cell detector; tritiated water; granulated scintillator

## 1. Introduction

Tritium, commonly found in the form of tritiated water, is an important radionuclide being used as a fuel in fusion reactors [1] and as a radioisotope tracer [2,3]. Tritium only emits low-energy beta radiation up to a maximum energy of $18.591 \pm 0.059$ keV [4], existing naturally in the environment as a product of the interaction of cosmic rays with the atmosphere. However, since the advent of nuclear technology, more significant quantities of tritium have been produced through the operation of nuclear fission reactors and subsequent waste operations and reprocessing, nuclear weapons testing [5], and particle accelerators [6], allowing its release into the environment. A recent example of such a

release was the Fukushima disaster, which, as reported in 2020, was producing 55,000 to 80,000 m$^3$ of tritiated water a year at an average activity of 1000 Bq/cm$^3$ [7]. Due to the presence of tritium in the environment, detection limits within drinking water have been put in place by the World Health Organization at a guidance level of $10,000$ Bq/L$^{-1}$ [8] and by the European Commission at 100 Bq/L$^{-1}$ under the Drinking Water Directive [9].

Tritium is incredibly difficult to detect as it is very chemically similar to hydrogen and produces no gamma rays, meaning it cannot be identified using any characterized photopeaks. Its emitted beta particles can only travel a small distance (less than 4.088 μm [10] in water) before being attenuated by the surrounding medium, most commonly water or air. Because of this, the vast amount of emitted radiation does not reach the detector and so is not measurable. This greatly lowers the overall detection efficiency of many detectors, requiring longer count times, higher activities, or larger scintillator surface areas. Liquid Scintillation Counting (LSC) is one method commonly used for the detection of tritium. Here, a liquid cocktail of light-producing molecules (the scintillator) is mixed directly with the sample, thus reducing the distance between the sample and scintillator to the molecular level, resulting in a much greater probability of interactions occurring than with traditional solid scintillators. The downside of such a method is that the mixed cocktail solution cannot easily be recovered or reused with another sample, and so this method results in a larger quantity of radioactive solution that must be handled and disposed of. Due to these factors, LSC has been restricted to laboratory analysis with few to no examples of such a system working in real time within an in situ environment.

Alternative methods of tritium detection using scintillators have been investigated in previous publications. For example, Azevedo et al. [11] used a scintillator (the type of scintillator used was not disclosed) formed of a 1 mm diameter fiber-bundle optically connected to two photomultiplier tubes (PMTs) in coincidence counting mode. Two prototypes were created and tested with 108.11 MBq/L tritiated water, showing an increased channel count when compared to the background.

Another flow-cell detector by Jun Woo Bae and team [12] used sheets of a plastic scintillator layered upon each other to form a rectangular cavity. The detector was tested with two concentrations of tritiated hydrogen gas, finding a clear improvement in the 12-channel chamber when compared to a single-channel chamber, with a detection efficiency of $1.78 \pm 0.04\%$ for the single channel and $27.91 \pm 0.49\%$ for the 12-channel chamber. Heterogeneous scintillators have also been investigated [10] using simulations to find that tritium has a maximum straight-line track length of 4.088 μm for its emitted beta particles within pure water and obtained an optimal scintillator particle radius of 10 μm for aqueous tritium detection. Finally, Kawano et al. [13,14] developed a tritium flow cell using a Teflon PHA tube detection cavity filled sequentially with 50, 100, and 300 μm diameter particles of CaF$_2$(Eu). CaF$_2$(Eu) was identified as ideal by the team due to its very good chemical stability and good luminescence. A recent review of radiometric techniques for the assessment of aqueous tritium [15] lists and discusses many other systems, including tritium pre-enrichment, that have been published in the realm of tritium detection.

## 2. Scintillator Configurations

Previous publications [10,13,14] have shown that granulated forms of CaF$_2$(Eu) have a reasonable efficacy in detecting tritium. The work presented here builds on these works and demonstrates how a finely powdered scintillator maximizes the active surface area and allows for good detection efficiency for a flow cell-based detector of limited size.

A granulated scintillator enhances some of the benefits of homogeneous solid scintillators by having a far higher surface area and allowing the liquid sample to pass and envelope the particulates instead of remaining at the face of a single solid crystal. Due to the relatively small geometric path length of betas emitted by tritium within solid scintillators (less than 1 μm in CaF$_2$(Eu) [16]), using a solid scintillator thicker than 1 μm would not be advantageous for increasing detection efficiency, as almost no interactions would occur at or over this depth. A needlessly thick scintillator would also increase the optical attenuation

of the scintillated photons to the PMT or silicon photomultiplier (SiPM), further reducing overall efficiency. This behavior is shown in Figure 1.

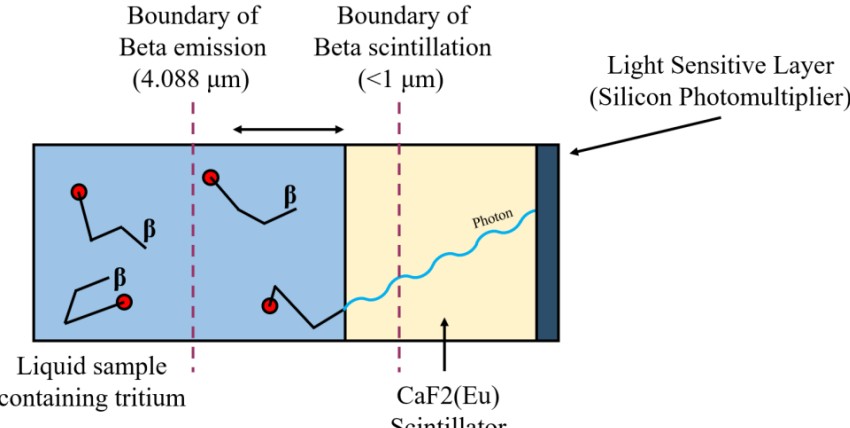

**Figure 1.** Diagram of a solid block of scintillator in contact with a tritiated liquid sample.

The main benefit of granulated scintillators, compared to a liquid scintillator, is that the particles can be secured while allowing a sample to flow through the detection cavity. This allows the scintillator to be reused until it is soiled, as excess surface contamination of the scintillator will increase beta attenuation, resulting in reduced detection efficiency. If using a liquid scintillator, a cocktail would have to be expelled with the sample and then replaced. For in situ environments, a large volume of scintillation cocktail would have to be stored to feed such a flow cell, whereas a granulated flow cell could be periodically replaced with another. It should also be noted that scintillation cocktails are often a hazard to the environment due to their toxic nature and high solubility in water. For example, GoldStar LT2, commonly used for low-energy beta and alpha detection, is very toxic to aquatic life with long-lasting effects [17]. Due to this hazard, any used sample mixture leaving the cavity would have to be stored for future removal and waste processing, while only the original sample would leave the granulated flow cell with sufficient filtering.

Granulated scintillators also have their downsides, principally that a granulated scintillator will have poorer optical properties compared to a conventional solid scintillator due to the number of boundaries each photon would have to pass in reaching the photomultiplier detector. Second, the poorer mixing with sample compared with a liquid cocktail due to the finite particle size. Therefore, for these reasons, the validity of these granulated forms has been investigated below.

*Granulation Method: CaF$_2$(Eu)*

The scintillator CaF$_2$(Eu) is a glass-like material and, therefore, can be easily ground into a granular form. This was performed using a mortar and pestle, as per a method used by Alton et al. [16], whereby approximately 20 g of rough-cut CaF$_2$(Eu) sourced from Advantech UK was manually crushed over 5 min. The resulting powder contained a wide distribution of particle sizes, and therefore, a second process was required to separate them.

The crushed power was added to a stack of sieves. The sieves had the following mesh sizes: 1000, 500, 355, 250, 125, 90 and 50 μm. These sieves were then placed into an auto sieve and left to shake for 10 min. Afterward, the powder from each sieve was collected and placed into separate sample containers.

## 3. Tritium Detection Using Powdered Scintillators

As finer powders are more difficult to contain in flow cells, there is a trade-off between the size of powders that can be reliably contained using filters, etc., and their ability to detect low-energy betas. To ascertain the optimal size, the detection efficiency of different particle sizes has been determined. A total of 5.00 ± 0.01 g of sieved CaF$_2$(Eu) powder

of a specific size range was added to a glass vial along with $1.80 \pm 0.01$ g of diluted tritium, producing a solution with an activity of 2079.5 Bq/g. A set of control vials was also produced with $1.80 \pm 0.01$ g of deionized water mixed with the powdered scintillator instead of the diluted tritium to determine background count rates. All vials were then placed in a Tri-Carb 3170TR/SL liquid scintillation counter (LSC) for 240 min each, and the counts were monitored between a lower energy limit of 2.0 keV and an upper energy limit of 18.6 keV. The resulting data collected by the LSC are shown in Tables 1 and 2 for active sample vials and background vials, respectively.

**Table 1.** Masses within sample vials containing Tritiated water for powder testing with the LSC. Weights of the scintillator added to each vial are within $\pm0.0001$ g. All measurements were repeated three times for 4 h each, and statistics were calculated. Background counts have been removed from the listed counts.

| Vial Number | Size Range (µm) | Detected Count (CPM, 3σ Error) | Powder Detection Efficiency % (3σ Error) |
|---|---|---|---|
| 1 | 500 to 1000 | $31.27 \pm 0.60$ | $0.01 \pm 0.00$ |
| 2 | 355 to 500 | $45.53 \pm 4.58$ | $0.02 \pm 0.00$ |
| 3 | 250 to 355 | $61.77 \pm 5.21$ | $0.03 \pm 0.00$ |
| 4 | 125 to 250 | $100.82 \pm 2.46$ | $0.04 \pm 0.00$ |
| 5 | 90 to 250 | $306.85 \pm 12.56$ | $0.14 \pm 0.01$ |
| 6 | 50 to 90 | $383.40 \pm 12.40$ | $0.17 \pm 0.01$ |
| 7 | 0 to 50 | $440.70 \pm 29.86$ | $0.20 \pm 0.01$ |

**Table 2.** Activity of non-active control samples denoting background activity. All measurements were repeated three times for 4 h each, and statistics calculated.

| Vial Number | Size Range (µm) | Detected Count (CPM, 3σ Error) |
|---|---|---|
| 8 | 500 to 1000 | $3.80 \pm 0.36$ |
| 9 | 355 to 500 | $3.77 \pm 0.23$ |
| 10 | 250 to 355 | $3.88 \pm 0.49$ |
| 11 | 125 to 250 | $3.91 \pm 0.69$ |
| 12 | 90 to 250 | $3.91 \pm 0.14$ |
| 13 | 50 to 90 | $3.80 \pm 0.53$ |
| 14 | 0 to 50 | $3.57 \pm 0.19$ |

A clear trend can be seen in Figure 2, showing that as scintillator particle size decreases, the count rate detected by the LSC increases, resulting in a higher detection efficiency when using a finer grid size. This can be attributed to the smaller particulates having a greater surface area between the sample and scintillator and, therefore, a greater probability of a beta particle entering and interacting with the scintillator. The peak detection efficiency for tritium detection was measured at $0.20 \pm 0.01\%$, for the size range of 0 to 50 µm, and so this sieve grid size was used to produce the powder for the tested laminated flow cells.

The count rates obtained for the finest powder group show an improvement over previously published results [13], which found for their 50 µm $CaF_2(Eu)$ granular scintillator a relationship of:

$$\text{CPM} = 0.1367 \times \text{Activity Concentration} \left( \frac{\text{Bq}}{\text{g}} \right) + 0.191 \tag{1}$$

Here, the concentration of tritium used in the vials was 2109.9 Bq/g, resulting in a count rate of $440.70 \pm 29.86$ CPM, while the relationship described in the previous publication formulates a count of approximately 288.61 CPM when calculated for the same concentration. This improvement could be attributed to the presence of finer particles and the more efficient counting setup of the LSC system used here. Notably, the size range of the scintillator had no clear effect on the background count rate, keeping to an average of $3.81 \pm 0.56$ CPM.

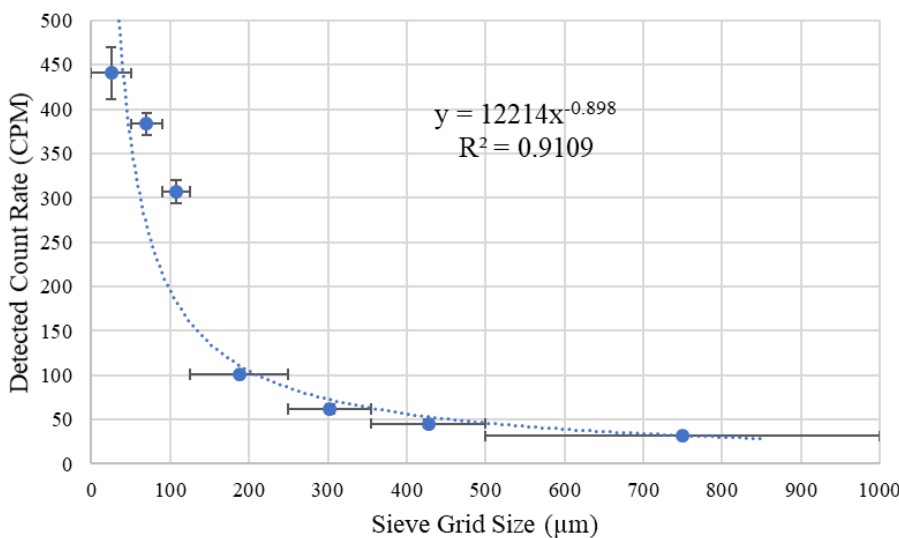

**Figure 2.** Graph of the resulting data from the counted vials containing mixtures of dilute tritium and CaF$_2$(Eu) scintillator. A power trendline (dotted blue line) has been added to the plot along with its R$^2$ value and trendline equation.

## 4. Laminated Flow-Cell

A flow cell is a cavity in which a liquid sample can be passed through to take measurements. Here, a new form of flow cell utilizing heat-based lamination has been implemented. In this process, two sheets of plastic are adhered along their edges to one another using heat, forming a watertight seal. The benefit of this over a more conventional block/rectangular cavity is that many laminated flow cells can be made rapidly with minimal equipment, with a single flow cell requiring 30 min of cutting and laminating followed by two days for the adhesive to cure. The shape created by lamination is suited to coincidence counting as there are two flat transparent sides through which scintillated photons can pass. These sides can then be placed in contact with SiPMs or PMTs to detect said photons.

A cross-sectional diagram of the flow cell can be seen in Figure 3, labeled with its key components. Three example beta decays are also included, showing how, after entering the scintillator, resulting photons can travel outwards toward the two arrays of SiPMs, which are isolated from the liquid solution via the laminated sheets.

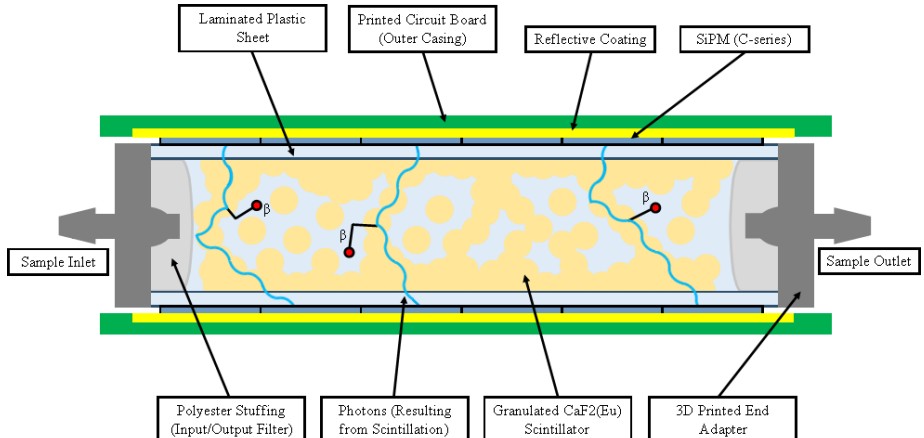

**Figure 3.** Cross-sectional diagram of a laminated flow cell with the outer casing. Within the cavity, three beta decays have been added, showing paths photons take to reach both SiPMs, not to scale.

The laminated cavity was filled with the powdered scintillator produced in the prior experiment discussed in Section 3. The size range of particulates under 50 μm was selected due to it having the highest detection efficiency compared to the other size groups tested.

Polyester filter wool is added to both ends to ensure the retention of the scintillator powder during operation. The outer casing compacts the filter material, improving its ability to retain the powder. Two stereolithographic (SLA) printed end adapters allow standard clear silicone tubing to be connected to both sides of the cavity. The adapters have been designed with a ridge for the ends of the laminated envelope to slip into (See Figure 4). Adding low-viscosity glue allows a watertight connection with the flow cell.

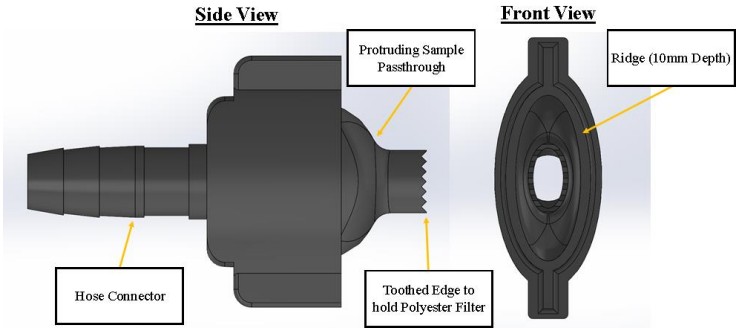

**Figure 4.** Side and front view of 3D printed end adapter labeled with key design features.

Scintillated photons produced in the flow cell are detected by SiPMs. SiPMs have been selected as they can be configured in compact arrays. In this case, a column of SiPMs has been placed along the length of the flow cell. This configuration is readily extendable. The greater the length of the cavity, the more SiPMs can be integrated, resulting in an increase in the total surface area between the sample and scintillator and an increased light-detection area.

Another reason for the use of SiPMs over PMTs is their small thickness of 0.65 mm [18], around 29 times thinner than a comparably sized square PMT (R11265U series [19]) and around five times thinner than the smallest PMT brought to market (R12900U series [20]). The thinness of the sensor greatly affects the overall diameter/thickness of the flow-cell assembly, and therefore, SiPMs will allow the use of the detector in much more constrained applications like within thin pipes and boreholes. Finally, SiPMs are safer to operate when placed within proximity to water as they function using lower bias supplies, operating off tens of volts as opposed to thousands of volts, lowering the risk of electric shock.

Here, the length of the cavity was approximately 18 cm end to end. The process of producing the flow cell is as follows, with the steps depicted in Figure 5.

1.  Two rectangular laminate sheets are cut, measuring 18 cm by 3.5 cm.
2.  The sides of the sheets are heat-sealed together, leaving an opening at each end. The excess sheet material is trimmed.
3.  One end of the laminated envelope is pinched shut, and approximately 5 g of granulated scintillator is added, followed by polyester filter wool into both ends.
4.  Plastic adapters are bonded onto each end of the filled flow cell, creating an enclosed cavity.

To pair with the flow cell, an outer casing was devised to improve the overall rigidity of the cavity as well as to provide an anchor point for multiple SiPMs to run alongside the laminated faces. A custom printed circuit board (PCB) allowed the selective placement of a strip of solder pads compatible with the Onsemi C-series line of SiPMs as well as un-masked "copper pours" which improve the inner reflectivity of the casing, acting as mirrors reflecting stray photons into the cavity.

Multiple 2M screw holes were also included to allow even pressure across the whole length of the flow cell. Spacers made from small strips of PCB limited the thickness of the cavity to approximately 4.8 mm while also helping to reduce light leakage entering from the edges. It was noted while testing the system that the pressure applied by the casing had a large effect on the flow rate through the cavity and that making the casing too loose would allow scintillator powder to migrate into the polyester stuffing, reducing detection efficiency. The addition of foam around the polyester filters helped to pinch the

ends around the scintillator, keeping it within the center of the cavity once the casing was in place.

For the final iteration of the casing design, thermal vials, and external copper pours were added around the SiPMs to improve the temperature regulation from the cooling system (discussed in Section 5.1) by reducing the thermal resistance across the casing walls. Discussion and modeling of thermal bias show a clear relationship between the number of vials and the thermal resistance across a simulated square PCB board [21]. PCBs utilizing thermal vials have also been tested with liquid cooling systems [22], similar to the cooling system that will be discussed in this work. Figure 6 shows labeled images of the designed and tested flow cell both before and after assembly of the outer casing.

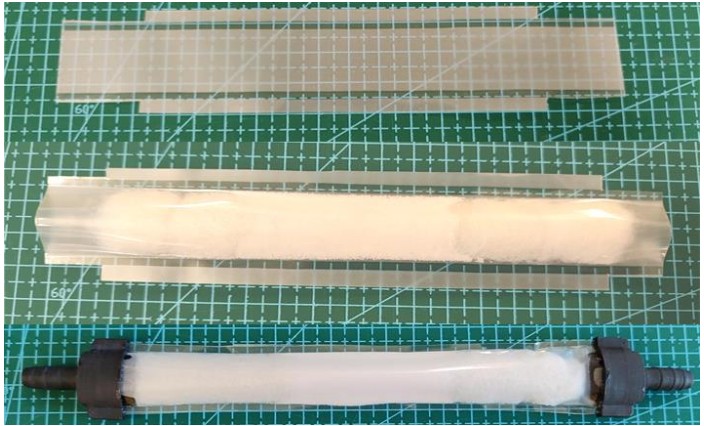

**Figure 5.** Images showing a flow cell being manufactured: (from top to bottom) sheets are cut and laminated, filled with granulated scintillator, and end adapters attached.

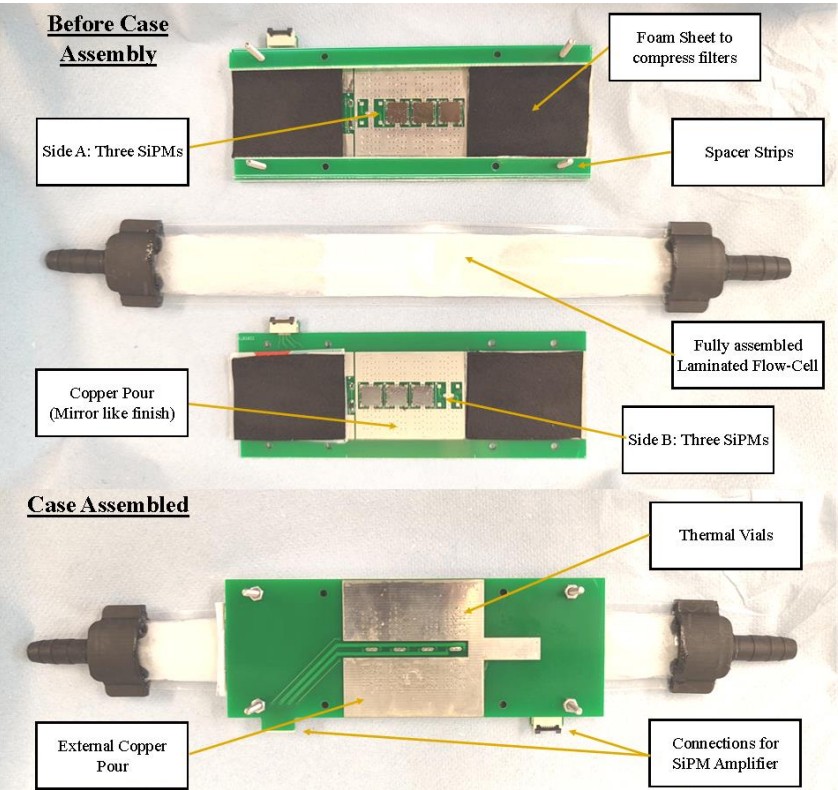

**Figure 6.** Image of the flow cell and casing parts both before and after assembly of the outer casing around the laminated cavity. Two FFC connectors allow rapid connection and disconnection of each SiPM array from the electrical system.

## 5. Detector Setup

For the backend electrical system, a previously developed system has been used [23] with some improvements to configurability. The system (shown in Figure 7) is made up of three key parts: a two-channel data acquisition board (DAQ), two independent SiPM amplifiers, and two sensor heads. In addition, spectrum data processed by the DAQ is sent to a single-board computer (Raspberry Pi Zero W) so that the data can be stored within a database for future access. The system can create energy spectra based on pulses emitted by SiPMs when scintillated photons reach both SiPM arrays. Each SiPM array has its own SiPM amplifier board and analog-to-digital converter, which from now on will be referred to as a "channel".

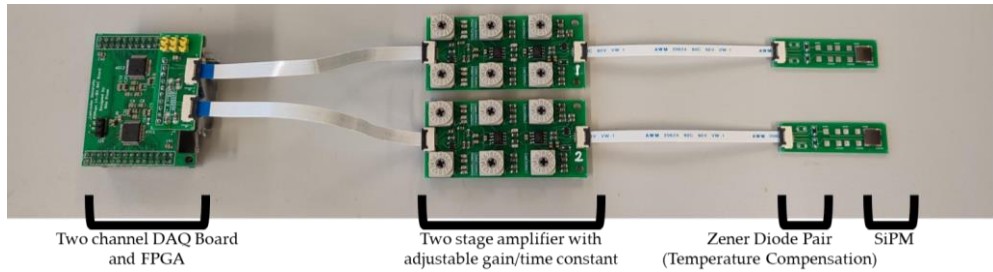

**Figure 7.** Image of an electrical system used in the flow-cell detector. Two channels perform coincidence counting with the SiPM sensor heads placed across the center of the flow cell. Temperature-stabilizing Zener diodes have been placed near the SiPMs to ensure good temperature matching.

Additional rotary switches added to the amplifier boards allow the gain and time constants to be configured easily in testing with 4096 possible settings. These settings were varied to maximize the spectral range for tritium decay while keeping noise to a minimum. Ribbon cables have been used with matching connectors to allow parts to be easily swapped, reconfigured, and replaced. A total of six SiPMs have been used—three connected in parallel for each channel.

As both sides of the flow cell are transparent, SiPMs can be added to both sides for coincidence counting. Other detectors for tritium have also previously implemented coincidence counting, most commonly using photomultiplier tubes [11,14,24–27]. Coincidence counting reduces false counts caused by noise in the electronics and dark pulses from the SiPMs or PMT, which would overwise overshadow counts from tritium decay. Coincidence counting using SiPMs has also been successfully implemented, demonstrating the suppression of dark noise to 1.69% of that compared to non-coincidence counting [28].

As the SiPMs can easily be saturated by external light sources, the flow cell, sample beaker, pump, amplifiers, and DAQ board were placed within a lightproof container with sealed panel mount pass-throughs for coolant, serial communication, and power.

The flow cell forms a closed loop using silicone tubing and a reservoir containing the sample to be analyzed, with a peristaltic pump added to the outflow side of the flow cell to pump the sample around the loop. The reservoir allows the concentration of tritiated water to be changed between measurements, as well as facilitating the flushing out of the flow cell after the experiment. The flow cell is operated under negative pressure to prevent leaks and to help force the sample through the scintillator powder. A diagram of the experimental detector setup can be seen in Figure 8.

The power supply for the DAQ board, single-board computer, SiPM bias, and amplifiers has been placed externally to reduce heat buildup within the enclosure; a single cable runs from the power supply unit to the detector, providing all electrical connections. Four separate supplies were required for 3.3 V, 5 V, 12 V, and −30.2 V. Excluding the pump and single-board computer, the electrical system draws approximately 250 mA, with the single-board computer pulling an additional 250 mA.

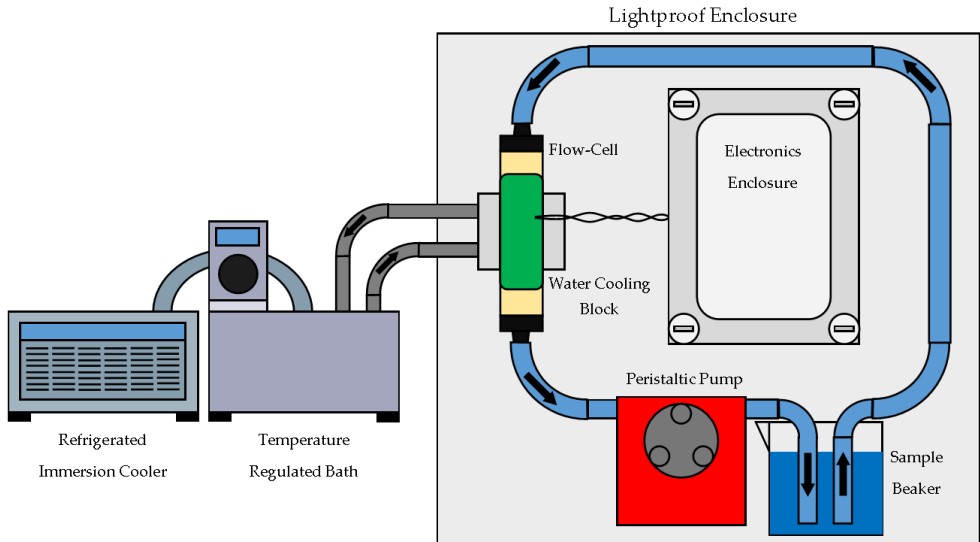

**Figure 8.** Diagram of detector setup: key parts include the lightproof enclosure, silicon tubing, electronics enclosure, flow cell, flow-cell casing, peristaltic pump, and sample beaker. The cooling bath and cooling pump were placed directly next to the enclosure.

### 5.1. Cooling System

As counts from the detection of tritium will be received in conjunction with thermal noise from both SiPMs, it is important that the temperature of the detector system is kept constant, or the level of noise will vary across measurements, leading to inconsistent results. A previous study [29] looked at the signal-to-noise ratio (SNR) of a PMT versus a SiPM cooled using a Peltier cell. They found that the SNR of the SiPMs would rise to that of the tested PMT at a temperature of 3 °C, implying that to achieve comparable noise levels to existing PMT-based tritium flow cells, a cooling system for the SiPMs would have to be added.

The solution implemented uses two hollow aluminum water blocks (40 mm height, 40 mm width, and 12 mm thickness), which are placed and fixed around the outside of the flow-cell casing. Coolant comprised of a mixture of 20% glycol ethylene to 80% deionized water was constantly passed through both heatsinks as the detector was running using the inbuilt pump of a Grant TX150 at pump speed 1. This coolant was kept at a constant temperature of 4 °C using the Grant-regulated liquid bath combined with an immersion cooler.

After a preliminary round of testing with this system, it was found that the SiPMs were becoming damaged. This led to the realization that condensation building up within the inside of the flow-cell casing would pool around the SiPMs, leading to moisture damage. Although this damage could not be often identified visually, placing an oscilloscope on the output of the SiPM amplifiers would show greatly reduced electrical background noise (from 100 mV-pp to below 20 mV-pp) when damaged and a reduced/no sensitivity to light. As a result, an acrylic anti-corrosion coating was selectively added to the inner sides of the casing, with a total of three coats used. Before its addition, the top faces of the SiPMs were covered with glossy plastic tape to stop the coating from adhering to light-sensitive faces directly. This cover was then removed partway into drying.

To further limit condensation and buildup of water droplets, 3 mm thick foam was adhered to the open faces of the cooling blocks, and the set temperature of the cooling system increased from 3 to 4 °C. Since these changes, the detector has been successfully tested for multiple days continuously.

### 5.2. Temperature Effects on Background Count Rate

To investigate how large of an effect temperature has on the SiPM output measurements, a continuous background reading of the flow cell filled with deionized water was

measured as it was cooled down from room temperature (22.1 °C) to 4 °C. The time taken to reach its set temperature was approximately 30 min.

The results are shown in Figure 9, where a clear exponential trend from 10 CPM of background counts at room temperature to 0.1 CPM at 4 °C can be seen, signifying that as the SiPMs are cooled, the false counts introduced as the result of noise are greatly reduced.

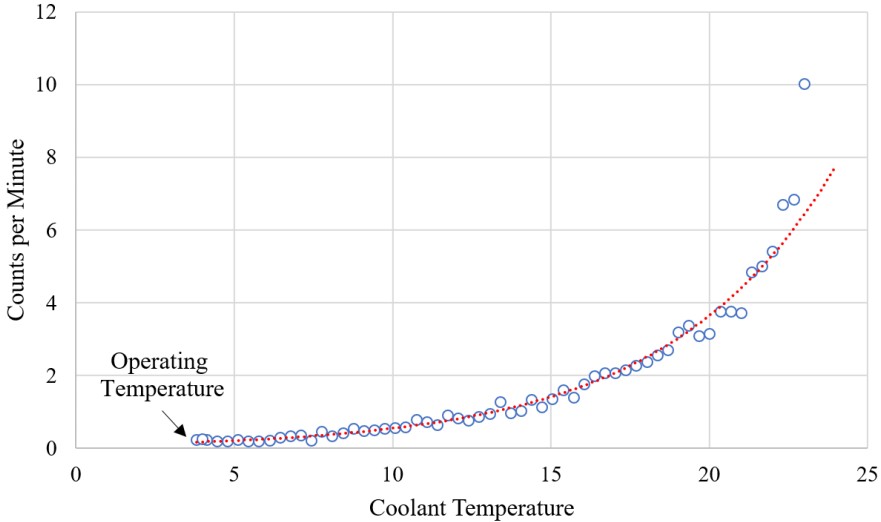

**Figure 9.** Graph plotting the total counts per minute recorded and measured over the cooling period. An exponential fit (dotted red line) has been added to the data points.

## 6. Flow Cell Detection of Dilute Tritiated Water

To validate that the flow-cell system is capable of low-energy beta detection, the system was tested by passing multiple concentrations of dilute tritiated water through the cavity in turn while recording the spectrum and count rate. The sample of tritiated water used had an activity concentration of 2058.08 Bq/g, and the volume of flow cells, filters, and piping was approximately 11 mL.

The experiment methodology is as follows: the flow-cell system was cooled and kept at a constant temperature of 4 °C, and a spectrum of background activity was first measured for 23 h with the granulated cavity filled with 20.003 g of deionized water. Then, a measured mass (See Table 3) of tritiated water was added to the sample reservoir and left to mix with the deionized water and circulate through the flow cell for four to six hours prior to measurement.

**Table 3.** Measurement time, total mass of water, and dilute tritiated water present in each round of the experiment. Weighting scales were calibrated to within ±0.003 of a gram. Measurement time to within 21.48 s.

| Round No. | Total Water Mass (g) | Total Tritiated Water Mass Added (g) | Measurement Time (Hours) |
|---|---|---|---|
| 0 (Background) | 20.003 | 0 | 23.236 |
| 1 | 21.004 | 1.001 | 23.236 |
| 2 | 22.023 | 2.020 | 60.556 |
| 3 | 23.018 | 3.015 | 41.168 |
| 4 | 24.061 | 4.058 | 22.317 |
| 5 | 25.073 | 5.070 | 85.069 |

The detector was then left for multiple days (the exact measurement time is listed in Table 3 below) to record counts from the flow cell. Over this period, the pump was turned off to avoid electrical noise affecting the recorded count rate. This process was repeated a further four times, cumulatively adding diluted tritiated water for a total of five measurements at five different tritium concentrations.

By the last measurement, a total liquid sample of $25.07 \pm 0.02$ g was passing through the detector system with an activity of 416.18 Bq/g. Table 3 displays the masses added each round and the total measurement time. Table 3 shows key recorded data obtained from each experiment round.

Included in Table 4 is the detection efficiency for the estimated amount of activity contained purely within the detection cavity itself, i.e., excluding activity in the beaker, piping, and filters. This was obtained by assuming that the powder within the flow cell is packed randomly and loosely. Therefore, its primary porosity is approximately 41% [21]. When the flow cell was manufactured, $4.73 \pm 0.02$ g of scintillator was added. Given the density of the scintillator [30], the amount of sample within the powder can be estimated as:

$$V_{\text{Liquid Sample}} = \frac{0.41 \times \text{Mass}_{\text{Scintillator}}}{\rho_{\text{CaF2(Eu)}}} = 0.610 \text{ mL} \qquad (2)$$

**Table 4.** Detected CPM was recorded by the flow-cell detector for each round, along with the calculated full loop tritium activity concentration based upon the added mass of tritium after mixing.

| Round No. | Post-Mixing Tritium Activity (Bq/g) | Detected CPM (Background Removed) | Detection Efficiency of Activity in Cavity % (3) |
|---|---|---|---|
| 0 | 0 | $0 \pm 0.331$ | - |
| 1 | 98.082 | $3.682 \pm 0.674$ | $0.103 \pm 0.010$ |
| 2 | 189.753 | $11.245 \pm 0.475$ | $0.163 \pm 0.002$ |
| 3 | 269.574 | $14.609 \pm 0.546$ | $0.148 \pm 0.002$ |
| 4 | 347.123 | $16.866 \pm 0.728$ | $0.133 \pm 0.003$ |
| 5 | 416.180 | $22.759 \pm 0.443$ | $0.149 \pm 0.001$ |

Therefore, the detection efficiency of activity surrounding the scintillator ($\varepsilon$) can be calculated using the following equation [31]:

$$\varepsilon(\text{Estimated activity in cavity})_{\%} = 100 \times \frac{S - B}{60 \times A \times V} \qquad (3)$$

where S is the count rate of the sample (CPM), B is the count rate of background (CPM), A is the activity concentration (Bq $\text{mL}^{-1}$) and V is the volume of sample surrounding the scintillator (mL).

The readings from the detector show an increased count rate of 3.682 CPM after the tritium activity concentration in the flow cell increases from 0 to 98.082 Bq/g. Figure 10 plots the background spectrum taken by the detector before any tritium had been added, followed by Figure 11, which plots the spectra taken by the detector after each addition of tritium, the background counts have been removed.

The saturation of the ADC (Analog-to-Digital Converter) can be seen in Figure 10 as a peak at the far-right end of the spectrum. This is due to the high gain of the SiPMs amplifiers, which results in pulses with energies above that which can be recorded by the top register of the ADC. This peak is, therefore, not a photopeak but merely the sum of all the pulses caused by interactions that are more energetic than this scale.

Figure 11 shows the resulting spectra recorded by the detector for each tritium concentration. As expected, most counts occur at lower channel numbers, overlapping with the detector noise seen in the background measurement results (Figure 10). Due to the implemented detection algorithm discussed in a previous publication [23], in which shaped pulses must meet a threshold, channel counts start to decrease rapidly below approximately channel 500, as pulses start to be rejected as they can no longer be differentiated from background noise.

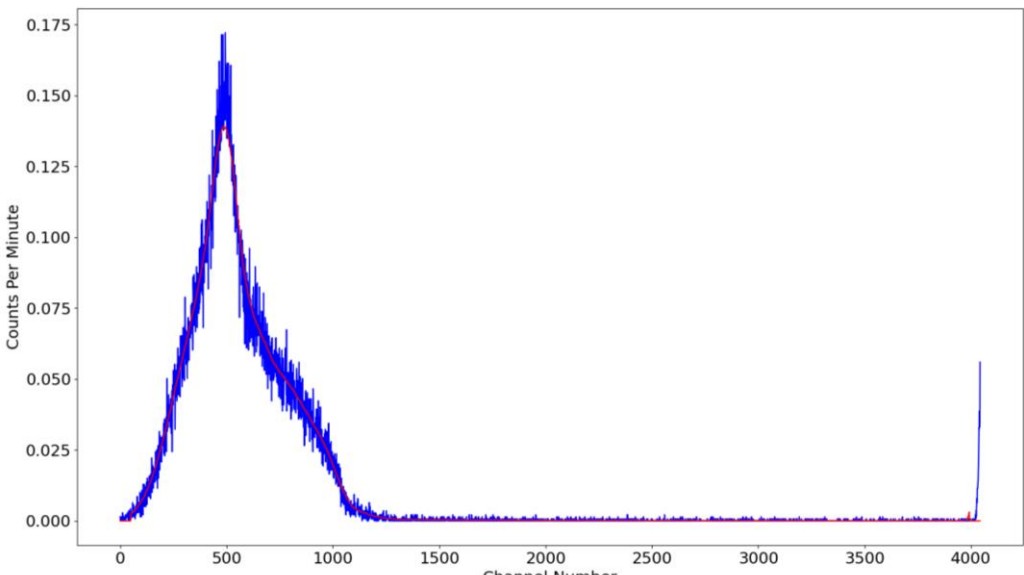

**Figure 10.** Spectrum of background taken by the flow-cell detector while filled with deionized water. The blue trace plots the raw channel data, and the solid red trace plots a 20-sample moving average. Spectrum was captured over 22.262 h.

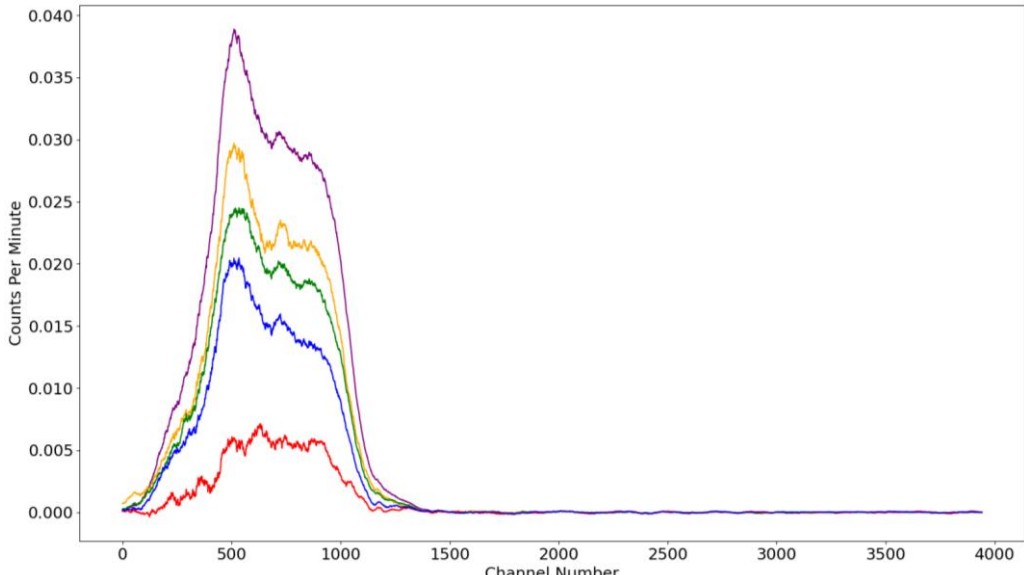

**Figure 11.** Spectra taken by a detector after each addition of tritium, background removed. A 50-sample moving average has been applied to each trace. Trace colors red, blue, green, orange, and purple represent the spectra taken in Rounds 1, 2, 3, 4, and 5, respectively.

A set error of $\pm 20$ Bq/g has been included in the plot. This has been added to represent the uncertainty in the activity within the flow cell due to factors such as the incomplete mixing of the added tritiated water, the movement of powder within the cavity, and air gaps being removed or introduced to the cavity as the sample is being passed. Interpreting Figure 12, a positive trend between tritium activity and counts per minute has been obtained. The added line of best fit provides a predicted behavior of:

$$\text{Tritium Activity(Bq/mL)} = 18.238 \times \text{Detector(CPM)} + 9.894 \tag{4}$$

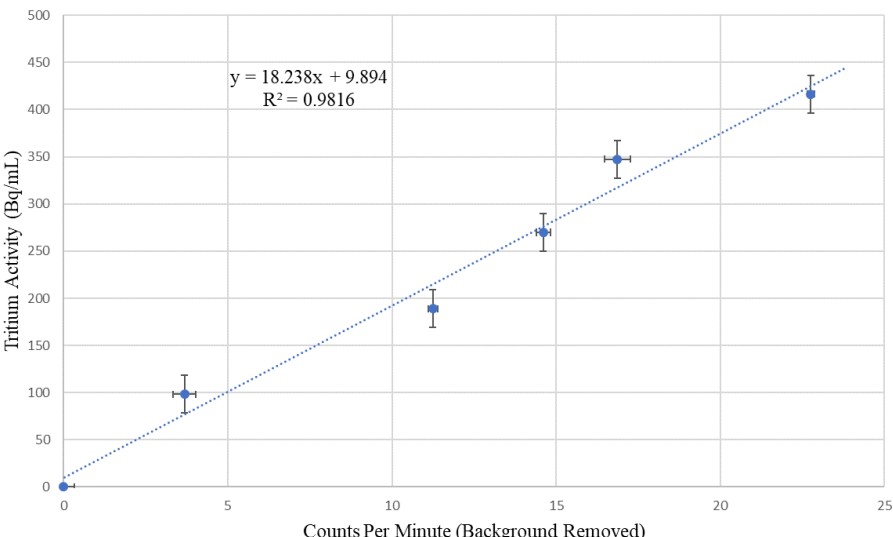

**Figure 12.** Tritium activity concentration within flow cell as a function of Measured CPM, where concentrations ranged from 0 Bq/g to 416.180 Bq/g. A linear line of best fit (dotted blue line) has been added to predict the detector's overall response to the activity of tritium.

For calculating the Minimum Detectable Energy (MDA), the following characteristic Equation (5) [31] has been used. The background count rate has been set as recorded CPM from the first round (round 0) of the flow-cell testing (58.861 CPM).

$$\text{MDA}(\text{Bq mL}^{-1}) = \frac{\frac{2.71}{t_s} + 4.65\sqrt{\frac{B}{t_s} + \frac{B}{t_b}}}{60 \times \varepsilon \times V} \tag{5}$$

If one day were allowed for measurement, the minimum activity concentration of tritium would be (including 3σ error):

$$\begin{aligned}\text{MDA}\left(\text{Bq mL}^{-1}\right) &= \frac{\frac{2.71}{24 \times 60} + 4.65\sqrt{\frac{58.861}{24 \times 60} + \frac{58.861}{23.236 \times 60}}}{0.01 \times 60 \times (0.139 \pm 0.005) \times 0.610} \\ &= 26.356 \pm 0.889 \text{ Bq mL}^{-1}\end{aligned} \tag{6}$$

## 7. Conclusions

As a result of this work, a flow-cell detector capable of the detection of tritium has been designed, built, and validated using multiple samples of dilute tritiated water. This system has also implemented a coincidence counting system based on SiPMs as opposed to the more commonly used PMTs, which, to our knowledge, is the first of its kind experimentally verified with tritium, therefore making this system novel for the detection of tritium. The use of SiPMs will allow much greater flexibility compared to PMTs, as the sensitive light-detection area can be easily configured to the elongated shape of the flow cell—in the case of this detector, in a strip. Implementation of SiPMs has also made the detector safer to operate in wet in situ environments due to the much lower required bias voltages.

The study of granulated scintillators within an LSC has shown that the method presented in this work can powder a solid block of $CaF_2(Eu)$ while allowing it to keep its scintillating properties. It has also demonstrated a relationship between size distribution and detected counts for tritiated water, obtaining a maximum detection efficiency of $0.20 \pm 0.01\%$ and count of $440.70 \pm 29.86$ CPM for $CaF_2(Eu)$ with particle sizes between 0 and 50 μm.

A total of 256 h of collected data from experiments with multiple concentrations of dilute tritiated water has validated the ability of the detector to detect low-energy betas with an average efficiency of 0.139% obtained from five rounds of measurements. An MDA value has also been obtained, finding that over a 24-h counting period, the minimum

concentration of tritium that can be detected is $26.356 \pm 0.889$ Bq mL$^{-1}$. Recorded spectra have also been obtained showing that counts introduced after the addition of tritiated water follow the expected energy distribution for tritium. These spectra also allow the possibility for other beta-emitting isotopes to be discriminated between in the future.

**Author Contributions:** Conceptualization, N.E.J.D., S.D.M., J.G. and D.C.; software, N.E.J.D.; validation, N.E.J.D.; formal analysis, N.E.J.D.; writing—original draft preparation, N.E.J.D.; writing—review and editing, S.D.M., J.G. and D.C; supervision, D.C., J.G. and S.D.M.; funding acquisition, D.C. and S.D.M. All authors have read and agreed to the published version of the manuscript.

**Funding:** This research was funded by the NDA PhD bursary scheme and EPSRC iCASE EP/X524797/1.

**Institutional Review Board Statement:** Not applicable.

**Informed Consent Statement:** Not applicable.

**Data Availability Statement:** The data presented in this work are available from the corresponding author.

**Conflicts of Interest:** The authors declare no conflict of interest.

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
