# Peer review of "Laminated Flow-Cell Detector with Granulated Scintillator for the Detection of Tritiated Water"

_radiation, doi:10.3390/radiation3040017_

Round 1

Reviewer 1 Report

Comments and Suggestions for Authors

A well-written instrumentation paper describes the usage of laminated flow-cell using granulated scintillator. The measurement was conducted comprehensively with appropriate data interpretation.

Please answer the following comments:

1. The usage of laminated flow-cell using granulated scintillator achieved a MDA (tritium) of 18.668 Bq/mL, which is much less sensitive in many orders of magnitudes than that of LSC technology (~1 Bq/L), which has been reported in many literatures. Please elaborate how this proposed technology could resolve the challenges that cannot be achieved with LSC?

2. The technology described in this paper uses sealed laminated flow-cell (by heat) with SiPM (dark noise required coolant) for tritium counting, in which, LSC requires only one-step preparation: a direct measurement of a mixture of cocktail and aqueous solution. Why should this technology that require multiple-step preparation without achieving better sensitivity be used?

3. Lines 61-62: The downside of such a method is that the mixed cocktail solution cannot easily be recovered or reused with another sample, and so this method results in a larger quantity of a radioactive solution.

Please explain how to reuse or recover the sample radioisotopes from the mixed granulated scintillator, particularly when they’re solidified? Also the proposed ratio of 5:2 of powdered scintillator to sample (described in Lines 141-142) is roughly the same ratio as the conventional LSC technology, thus similar volume of waste per measurement should be generated for both technologies.

Reviewer 2 Report

Comments and Suggestions for Authors

Please find some comments/suggestions to the revised version of your paper.

Reviewer 3 Report

Comments and Suggestions for Authors

This is a well-written paper that discusses research into an improved design for a water-flow-through radiation detector with a focus on detecting tritium in water. This is a topic of interest to a variety of applications studying tritium in industry and the environment.

This paper is appropriate for the journal and should be published with some minor edits.

There are a number of missing references in the paper that display as errors. These are on lines 99, 144, 145, 186, 201, 253, and 298.

The first two figures are labelled "Figure 1". This should be changed to a unique identifier, and subsequent figures labels should be modified.

Under section 5.1, it may be interesting to know why the authors did not utilize an inert atmosphere (e.g. from the boil off of liquid nitrogen) to mitigate the condensation issue.

The MDA calculation should be rechecked. The original reference is Currie,L.A.,1968. Limits for qualitative detection and quantitative determination. Application to radiochemistry. Anal. Chem. 40, 586e593. https://doi.org/ 10.1021/ac60259a007. The calculation of the MDA may need to be revised to the "paired observation" formula rather than the "well-known blank" formula if I understand the technique properly. That is, the mean of the background rate is similar to the variance, rather than the variance on the mean background being close to zero.
